# Recent Trends in Prevalence of HPV Infection in Nasopharyngeal Carcinoma in Japan

**DOI:** 10.3390/microorganisms13112514

**Published:** 2025-10-31

**Authors:** Luyao Liu, Nobuyuki Hirai, Satoru Kondo, Makiko Moriyama-Kita, Ryotaro Nakazawa, Shigetaka Komura, Makoto Kano, Daisuke Uno, Manabu Inaba, Takayoshi Ueno, Yosuke Nakanishi, Kazuhira Endo, Hisashi Sugimoto, Tomokazu Yoshizaki

**Affiliations:** Department of Otolaryngology-Head and Neck Surgery, Graduate School of Medical Science, Kanazawa University, Kanazawa 920-8640, Japan

**Keywords:** nasopharyngeal carcinoma, human papillomavirus, immunohistochemistry, non-endemic region

## Abstract

Nasopharyngeal carcinoma (NPC) is a malignant tumor in which the etiologic contribution of Epstein–Barr virus (EBV) is well established. However, similar to that of oropharyngeal carcinoma, some papers reported that human papilloma virus (HPV) contributed to the development of NPC in non-endemic regions. Previously, we conducted a study on HPV infection in patients with NPC between 1996 and 2015 in our department. The current study aims to evaluate the incidence and role of HPV infection in NPC pathogenesis using samples of NPC after 2015. Paraffin-embedded tumor samples from 26 patients with NPC who were treated at our department between 2015 and 2022 were analyzed. HPV polymerase chain reaction, p16 immunohistochemistry, HPV genotyping, and in situ hybridization for EBV-encoded RNA were performed to determine the viral infection status. Of the 26 patients, 19 (73%) were EBV-positive and HPV-negative, 1 (4%) was EBV-negative and HPV-positive, and 6 (23%) were negative for both EBV and HPV. The detection rate of HPV has slightly increased from 3% to 4% over the past decade. Although Japan is a non-endemic region for NPC, HPV infection is exceedingly rare and may have a limited role in NPC development in Japan. However, the detection rate of HPV has not significantly changed in the past decade, further supporting the view that HPV has a relatively small impact on the pathogenesis of NPC in Japan.

## 1. Introduction

Nasopharyngeal carcinoma (NPC) exhibits distinct geographical and ethnic distribution patterns [1], with a high incidence in specific populations, such as those in southern China and Alaska Natives [2]. Epstein–Barr virus (EBV) infection plays a pivotal role in the pathogenesis of NPC [3], although in recent years, the rate of positivity of EBV in patients with NPC has decreased in low-prevalence areas [4]. However, the incidence of human papillomavirus (HPV)-related oropharyngeal cancer has increased markedly in developed countries, including those in Europe, the United States [5], and Japan [6]. This decrease in the positivity rate of EBV might be related to the increase in that of HPV. While HPV has been associated with NPC development in non-endemic regions, research on HPV-associated NPC remains limited, and its pathological and epidemiological characteristics are not yet well defined [7].

Based on our previous research [8], we extended our data collection from 2015 to 2022 to explore whether an upward trend in HPV positivity in NPC could be observed, similar to that reported in oropharyngeal carcinoma. The epidemiological features of NPC in Japan are unique, including characteristics of both endemic and non-endemic regions. The overall incidence of NPC in Japan is low, similar to that in non-endemic regions [9]; however, in terms of histological subtype, World Health Organization (WHO) type II and III NPCs predominate, similar to the patterns observed in endemic areas. Additionally, although EBV infection remains the dominant etiological factor for NPC in Japan, HPV/EBV co-infection has also been identified in some cases [8], and we recently identified a case of NPC with HPV infection alone. These observations suggest that HPV may be involved in NPC pathogenesis regardless of EBV status. Elucidating this involvement is clinically important, as viral etiology may influence diagnostic strategies, prognosis, and opportunities for targeted prevention. Therefore, in this study, we aimed to further explore the possible involvement of HPV in the pathogenesis of NPC in Japan over the past decade.

## 2. Materials and Methods

### 2.1. Patients

This study included data pertaining to all consecutive patients diagnosed with primary NPC at Kanazawa University Hospital between July 2015 and May 2022. All patients were from the Hokuriku region. Twenty-six patients with paraffin-embedded tumor tissue samples were analyzed. All patients underwent comprehensive clinical evaluation, and TNM staging (based on the 8th edition of the American Joint Committee on Cancer (AJCC) Cancer Staging Manual) [10] was performed by assessing findings on nasopharyngoscopy, contrast-enhanced magnetic resonance imaging, computed tomography, and fluorodeoxyglucose positron emission tomography/computed tomography of the head and neck to assess the primary tumor, lymph node involvement, and distant metastasis. All tumor specimens were stained with hematoxylin and eosin, and the tumor was classified according to the WHO criteria (Table 1). This study was approved by the Kanazawa University Medical Ethics Committee (approval no. 2025-033). The need for written informed consent was waived, and patients were provided the opportunity to opt out.

### 2.2. Detection and Typing of HPV

#### 2.2.1. DNA Extraction

Paraffin-embedded tumor tissue samples from 26 patients were sectioned in 3 μm slices and placed in 1.5 mL nuclease-free tubes, five slices at a time. Genomic DNA was extracted using the Allprep DNA/RNA FFPE kit (Qiagen, Tokyo, Japan) and Deparaffinization Solution (Qiagen, Japan), following manufacturer’s instructions.

#### 2.2.2. HPV Polymerase Chain Reaction (PCR)

HPV DNA detection was performed using the universal primers GP5+ and GP6+ and the TaKaRa Ex Taq PCR kit (TaKaRa, Katsushika, Japan). Each 50 μL reaction mixture contained 500 ng of genomic DNA (quantified by NanoDrop), 5 μL of GP5+ primer, 5 μL of GP6+ primer, 0.25 μL of Ex Taq polymerase, 5 μL of 10× Ex Taq buffer, and 4 μL of dNTP Mix, with nuclease-free water added to reach a final volume of 50 μL. The polymerase chain reaction (PCR) cycling conditions were as follows: initial denaturation at 94 °C for 1 min; followed by 36 cycles of denaturation at 94 °C for 20 s, annealing at 48 °C for 30 s, and extension at 72 °C for 30 s; a final extension at 72 °C for 5 min; and storage at 4 °C. The amplified products were mixed with a loading buffer (TOYOBO, Osaka, Japan) and analyzed by 2% agarose gel electrophoresis using a 100 bp DNA marker (TOYOBO, Japan). Water was used as a negative control and UPCI-SCC-090, which was gifted by Dr. Robert Ferris (University of Pittsburgh), was the positive control.

#### 2.2.3. HPV Detection and Genotyping

Tumor samples that showed a 150 bp band in the HPV PCR assay were considered HPV-positive tumor DNA samples, and only these cases were used for further HPV genotyping. HPV genotyping was performed using the 21 HPV GenoArray Diagnostic Kit (Hybribio, Guangzhou, China) following the manufacturer’s instructions. This kit allows for the simultaneous detection and differentiation of 21 HPV genotypes, including 15 high-risk types (HPV16, 18, 31, 33, 35, 39, 45, 51, 52, 53, 56, 58, 59, 66, and 68) and six low-risk types (HPV6, 11, 42, 43, 44, and 81). PCR amplification of the extracted DNA was performed, followed by hybridization with specific probes using Hybribio’s patented HybriMem flow-through hybridization technology. The results were visualized using enzyme immunocolorimetry and interpreted according to the manufacturer’s guidelines.

### 2.3. Immunohistochemical Analysis

Paraffin-embedded primary tumor samples were sectioned in 3 μm slices for immunohistochemical analysis of p16 expression. In HPV infection, the E7 oncoprotein of HPV binds to the low-phosphorylated retinal membrane cell tumor protein pRb, and the negative feedback pathway through which active pRb inhibits p16 is blocked [11]. This leads to the overexpression of p16, which is therefore considered an indirect indicator of HPV-mediated oncogenic activity. Consequently, p16 is widely accepted as an alternative marker for HPV infection in oropharyngeal squamous cell carcinoma (OPSCC) according to the AJCC 8th edition staging system, its correlation with HPV status in other head and neck cancer subsites, including NPC, remains less clear [12]. Following antigen retrieval at 98 °C, the deparaffinized sections were treated with 10% hydrogen peroxide for 10 min to block endogenous peroxidase activity. The sections were then incubated with a protein blocking solution (Dako, Glostrup, Denmark) for 30 min, followed by overnight incubation at 4 °C with a p16 (JC8) primary antibody (Santa Cruz Biotechnology, USA). After washing three times with phosphate-buffered saline (pH 7.2), the sections were incubated with EnVision+ secondary antibody (Dako) for 30 min. Color development was performed using DAB (Dako) solution, and the sections were counterstained with hematoxylin. Samples in which more than 70% of cancer cells showed strong and diffuse nuclear and cytoplasmic staining were considered p16-positive.

### 2.4. EBV Detection

EBV status was evaluated using in situ hybridization (ISH) for EBV-encoded RNA (EBER) in paraffin-embedded tissue sections. The assay was performed using the PNA ISH Detection Kit and EBER PNA probe/fluorescein (Dako, Denmark), following the manufacturer’s protocol. EBER-ISH demonstrates high diagnostic accuracy for primary NPC, with a reported sensitivity of 98% and specificity of 100% [13]. Samples in which more than 90% of cancer cell nuclei exhibited distinct dark brown staining were considered EBV-positive.

## 3. Results

### 3.1. HPV Positivity

Based on the results of HPV PCR, only one (4%) of the 26 samples showed a positive result (Figure 1). This case was initially identified as an NPC, and HPV infection was confirmed by the presence of a 150 bp HPV-specific band.

### 3.2. HPV Status

Only one of the 26 samples displayed a clear amplification band. This sample was subsequently analyzed using the HPV 21-type genotyping assay, confirming the presence of HPV infection and genotype as HPV18 (Figure 2).

### 3.3. p16 Status

p16 immunohistochemical analysis was performed to further confirm the presence of HPV infection in the sample that showed a positive amplification band in PCR (Figure 3). Among the 26 samples, only two demonstrated strong p16 positivity (Table 2). One of the two p16-positive cases was also HPV DNA-positive.

### 3.4. EBV Status

EBER-ISH was performed to evaluate the association between EBV infection and NPC. EBER expression was identified by a distinct dark brown nuclear staining (Figure 4). Among the 26 samples, 73% were EBER-positive, and 27% were EBER-negative (Table 2).

## 4. Discussion

Subsequent to our previous departmental report [8], we analyzed data pertaining to patients with NPC who were treated at our institution between 2015 and 2022, to further explore the potential roles of EBV and HPV in NPC pathogenesis and changes in the prevalence of HPV positivity in NPC in this decade. In endemic regions, NPC is predominantly associated with EBV infection [14], and HPV detection rates remain low [15]. In contrast, the proportion of HPV-positive NPC tends to be higher in non-endemic areas, primarily affecting the oropharynx [15,16], with a particular association with WHO type I (keratinizing) squamous cell carcinoma [17].

Japan is considered a non-endemic area for NPC [8], and the majority of cases are EBV-positive [18]. However, some cases of HPV-positive NPC have also been reported [19]. Notably, in our cohort, the histological subtypes of HPV-positive NPC were predominantly WHO type II or III, which differs from the keratinizing type (WHO type I) commonly reported in HPV-positive NPC in non-endemic Western countries [20]. This may suggest the existence of ethnicity- or region-specific pathogenic mechanisms.

In this study, we analyzed 26 cases with NPC, among which only one case tested positive for HPV (genotype 18). This case was EBER-negative and classified as WHO type II. Compared to the study by Kano et al., which reported an EBV positivity rate of 86% in 59 cases, our study showed a positivity rate of 73%, indicating no substantial change in the prevalence of EBV-associated NPC in non-endemic regions of Japan. EBV remains the predominant etiological factor.

Previously, we reported a 3% HPV positivity rate [8], while in this study, we observed a slightly higher rate of 4%. Detailed information comparing the previous and current cohorts, including study period, total cases, region, institution, detection methods, HPV-positive cases, HPV genotypes, and EBV/HPV co-infection cases, is summarized in Appendix A Table A1. Despite this difference, both studies support the view that HPV plays a limited etiologic role in NPC. The incidence of HPV-associated NPC remains significantly lower than that of EBV-associated NPC. Importantly, we did not observe any cases of EBV and HPV co-infection in our cohort, consistent with previous reports suggesting a mutually exclusive relationship between these two viral agents [20].

Moreover, our study found that p16 immunohistochemistry did not always correlate with HPV status. Some cases that were both EBV- and HPV-negative showed p16 positivity, often with focal or atypical staining patterns. This suggests that p16 expression alone is not suitable for a surrogate marker of HPV infection in NPC and should be interpreted in combination with HPV nucleic acid testing. In endemic areas of NPC, loss of heterozygosity in 9p chromatin, where p16 gene locates, is commonly observed in the NPC tumor [21]. The discrepancy of p16 detection between NPC and oropharyngeal carcinoma (OPC) could be attributable to the chromatin status [22].

Notably, although the incidence of HPV-associated oropharyngeal carcinoma has increased globally [23] and in Japan [6], our findings, together with previous reports, indicate that the incidence of HPV-associated NPC in Japan, particularly in the Hokuriku region, did not show an increased trend in the preceding decade. The result is in accord with a prior study finding no definite relationship between NPC and HPV [2]. These findings suggest that HPV-associated NPC remains uncommon in Japan, with a relatively stable epidemiological profile. This is an important observation of this study, indicating that HPV plays a small role in the pathogenesis of NPC in Japan.

The number of cases in this study is limited, and the findings represent data from a single institution in Kanazawa and may not represent the population in the remainder of Japan. To further clarify the clinical and molecular characteristics of HPV-associated NPC in non-endemic areas of Japan, and to determine whether this entity represents a biologically distinct subtype, we will initiate multi-center prospective studies with a long-term follow-up, larger numbers of cases, and integrative genomic and transcriptome analyses.

## Figures and Tables

**Figure 1 microorganisms-13-02514-f001:**
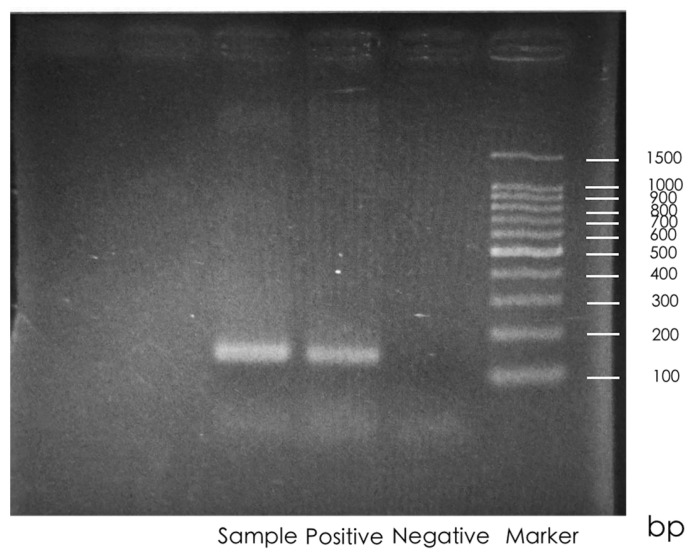
Representative result of gel electrophoresis for HPV. A 100 bp DNA marker was used for size reference. The first lane to the right of the marker is the negative control (no band). The second lane is the positive control (150 bp band). The third lane represents the test sample with an HPV-specific band at ~150 bp. HPV: human papillomavirus.

**Figure 2 microorganisms-13-02514-f002:**
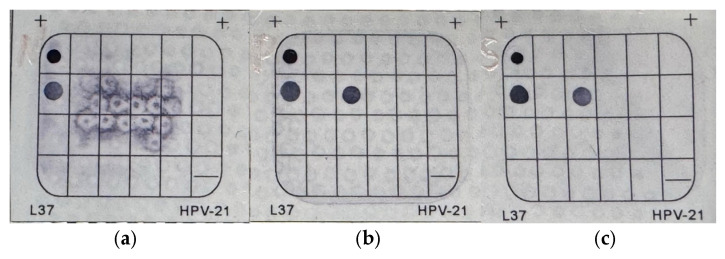
In each strip, the first row and the first column contain the biotin dot, serving as a system positive control to confirm the proper functioning of the color development system. The second row and the first column contain the IC dot, which ensures sufficient quality of the extracted DNA and successful PCR. (**a**) Negative control: only the biotin and IC dots are visible. (**b**) Positive control: provided by the kit, showing signals for both biotin and IC, along with a positive result for HPV type 18. (**c**) Sample: shows the same pattern as the positive control, indicating that the sample is positive for HPV type 18. IC: internal control; HPV: human papillomavirus.

**Figure 3 microorganisms-13-02514-f003:**
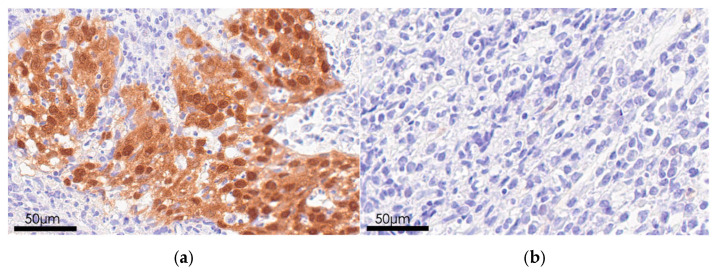
Immunohistochemical staining for p16. (**a**) Brown staining indicates p16 expression in the nuclei and cytoplasm of tumor cells in a patient with nasopharyngeal carcinoma. (**b**) The absence of diffuse brown staining in both the nuclei and cytoplasm of tumor cells was considered indicative of a negative result.

**Figure 4 microorganisms-13-02514-f004:**
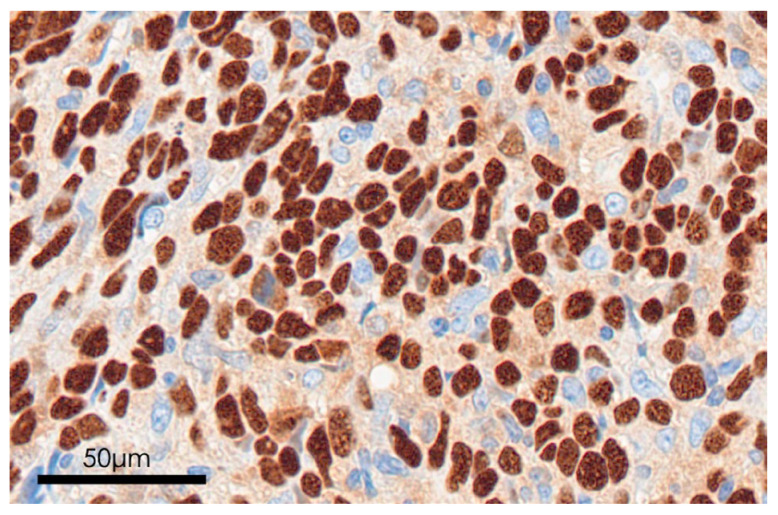
EBV-encoded RNA in situ hybridization. Dark brown staining indicates that EBV is expressed in the nuclei of tumor cells in patients with nasopharyngeal carcinoma. EBV: Epstein–Barr virus.

**Table 1 microorganisms-13-02514-t001:** Characteristics of patients according to viral status.

Characteristics	All Patients (n = 26)
Age	58.46 ± 12.79
Sex (Male/Female)	20/6
WHO type ^1^ (I/II/III)	1/20/5
Tumor classification (T1–T4)	8/10/5/3
Nodal classification (N0–N3)	2/11/11/2
Metastasis classification (M0/M1)	25/1
AJCC stage (I–IVB)	1/8/11/6/5/1
Treatment (RT ^2^/Alternating CRT ^3^/PCE-CCRT ^4^)	1/24/1

^1^ WHO: World Health Organization; ^2^ RT: Radiotherapy; ^3^ Alternating-CRT: Alternating cisplatin and 5-FU chemo-radiotherapy; ^4^ PCE-CCRT: Induction chemotherapy with Paclitaxel, Carboplatin and Cetuximab.

**Table 2 microorganisms-13-02514-t002:** Distribution of patients by EBER and p16 expression status.

Group	All Patients (n = 26)
EBER*+/p16−	19 (73%)
EBER−/p16+	2 (8%)
EBER−/p16−	5 (19%)

* EBER: Epstein–Barr virus-encoded RNA.

## Data Availability

All data supporting the findings of this study are included in the published article. Additional data, such as individual patient records, are available from the corresponding author upon reasonable request but are not publicly available due to privacy and ethical restrictions.

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
