# Peer review of "Recent Trends in Prevalence of HPV Infection in Nasopharyngeal Carcinoma in Japan"

_microorganisms, 2025, doi:10.3390/microorganisms13112514_

Round 1

Reviewer 1 Report

Comments and Suggestions for Authors

The authors explored an important topic in oncology, where the available data are limited, and studies like this manuscript are not as well-researched. Despite the small sample size, the study reveals very interesting insights to advance our understanding of a malignant neoplasm with a very aggressive development for patients. The details of the methodology are clear and well-designed. The figures illustrating the work are of excellent quality, adding to the submission's merits. The references used are from groups recognized for their familiarity with the topic. The discussion is consistent with the qualities previously mentioned.

Author Response

Dear reviewer1:

    We sincerely thank the reviewer for the positive and encouraging feedback on our manuscript.

Reviewer 2 Report

Comments and Suggestions for Authors

Review of the manuscript titled

Recent Trends in Prevalence of HPV Infection in Nasopharyngeal Carcinoma in Japan

by Luyao Liu et al.

         Nasopharyngeal cancer (NPC) varies geographically and ethnically.  It occurs especially in inhabitants of China and Alaska. Epstein-Barr virus infection plays a key role in the pathogenesis of NPC. However, in recent years, the rate of positive EBV tests in patients with NPC has decreased, especially in areas with low prevalence of this cancer. Japan is considered a non-endemic area for NPC, and the majority of cases are EBV-positive.

         The presented study aims to evaluate the incidence and role of HPV infection in

NPC pathogenesis. The authors examined 26 patients diagnosed with primary NPC at Kanazawa University Hospital in Japan between July 2015 and May 2022 and obtained the following results:

19 (73%) EBV-positive and HPV-negative, 1 (4%) - EBV-negative and HPV-positive, and 6 (23%) - negative for both EBV and HPV. They observed that the detection rate of HPV has slightly increased from 3% to 4% over the past decade. Moreover, they found that p16 immunohistochemistry did not always correlate with HPV status.

The manuscript is  well written. The research methodology is unquestionable. The tables and figures are unobjectionable. The authors based their work on 22 literature items.

The limitation of these studies is the small number of cases from one centre, which may not be representative of the entire Japanese population.

The observed trends are interesting. However, this requires confirmation in a much larger group so that the results can be generalized. Therefore, I believe that as preliminary research they deserve to be published.

Author Response

Dear reviewer2:

    We would like to sincerely thank the reviewer for the thorough evaluation and constructive comments on our manuscript.
    We fully agree with your observation that the limited number of cases from a single center is a potential limitation. We acknowledge this point and will address it in our future studies by including a larger multicenter cohort to confirm and expand our findings.
    Thank you again for your valuable and encouraging feedback.

Reviewer 3 Report

Comments and Suggestions for Authors

The manuscript by Lie et al describes a study of HPV prevalence in nasopharyngeal carcinoma in Japan. The study is relatively small (n=26) but appear methodically diverse. However, some uncertainty in methods description leaves room for problems.

Detailed comments to be addressed are listed below in page/line format

P2 L61 it is not completely clear whether the 26 patients were the complete set of patients treated or were those 26 patients selected for inclusion (ie had tissue remaining for analysis, consent, complete medical data…).

P3 L82-93 was some internal control amplification used? Ie beta-globin? Was some other means of assessing FFPE DNA degradation used? Issues might mean some samples were invalid instead of negative.

P3 L95-96 what is the amplicon size for GenoArray assay? It is relevant when performing PCR on FFPE. Was this assay validated on FFPE degraded DNA previously?

P3 L107 „p16 is widely accepted as an alternative marker for HPV infection.” The sentence is somewhat misleading since p16 is accepted as alternative for oropharyngeal site (acknowledged in AJCC 8th ed staging guidelines) but the correlation is less clear in other HNC subsites which is not suitably acknowledged in this sentence. Ie  https://pubmed.ncbi.nlm.nih.gov/32940116/

P5 L159. The results do not mention whether the DNA positive case overlaps with the two p16 positive cases?

P6 L175 the Table 2 inconsistently presents p16 and HPV DNA results. The total is currently 27, while it is likely that PCR results should be included in the numbers above

P6 L198 the discussion should include some additional info about previous study (ref 8) ie number of samples, how similar were populations then and now, which types were noted previously, was the method used the same. Also strongly consider adding the supplementary excel table combining the old and new cases as another useful reference about NPC). Both studies were individually small, but combined might represent a worthwhile resource

P6 L199 “4% positivity” is inconsistent with the “p16 is widely accepted as an alternative marker for HPV infection” statement (P3L107) since only DNA was taken as HPV positivity (1/26 =3.8%)

Typos/trivial issues

Throughout the manuscript - spacing around reference brackets.

P1 L36, no punctuation „areas[4] However,“

P2 L72 The treatment row uses the CCRT abbreviation which has no cases and can be omitted for brevity.

P4 L132 while irrelevant, it is customary to present the PCR electrophoresis images with the gel wells at the top and the shortest ladder fragments at the bottom (ie as the company selling the ladder does https://www.toyobo-global.com/products/lifescience/products/clon_008/index.html)

P4 L 142 „HPV 18-type genotyping assay” typo… should be HPV21-type genotyping assay

P5 L159 and L169. p16/EBER positivity is shown on Table 2 while text calls out to Table 1

Author Response

please read the attachment
